# Associations between Bovine Coronavirus and Bovine Respiratory Syncytial Virus Infections and Productivity, Health Status and Occurrence of Antimicrobial Resistance in Swedish Dairy Herds

**DOI:** 10.3390/antibiotics10060641

**Published:** 2021-05-27

**Authors:** Anna Duse, Anna Ohlson, Lena Stengärde, Madeleine Tråvén, Stefan Alenius, Björn Bengtsson

**Affiliations:** 1Department of Animal Health and Antimicrobial Strategies, National Veterinary Institute, SE-751 89 Uppsala, Sweden; info@annasdjurhalsa.com; 2Växa Sverige, P.O. Box 30204, SE-104 25 Stockholm, Sweden; anna.ohlson@vxa.se; 3Växa Sverige, P.O. Box 814, SE-391 28 Kalmar, Sweden; lena.stengarde@vxa.se; 4Department of Clinical Sciences, Swedish University of Agricultural Sciences, SE-750 07 Uppsala, Sweden; madeleine.traven@slu.se (M.T.); stefan.alenius@slu.se (S.A.)

**Keywords:** antimicrobial resistance, BRSV, BCoV, dairy cow, health, productivity, disease prevention

## Abstract

Bovine respiratory syncytial virus (BRSV) and bovine coronavirus (BCoV) affect dairy herds worldwide. In this study, effects on herd health, morbidity, and antimicrobial resistance (AMR) were assessed. Herds were considered free of infection (FREE), recently infected (RI) or past steadily infected (PSI) based on antibody testing of milk from primiparous cows. Data from farm records, national databases, and AMR of fecal *Escherichia coli* from calves were used as outcome variables. Compared to BRSV FREE herds: BRSV PSI herds had significantly higher odds of cough in young stock, a higher proportion of quinolone-resistant *E. coli* (QREC), but a lower proportion of cows with fever. BRSV RI herds had significantly higher odds of diarrhea in calves and young stock, a higher proportion of QREC and higher odds of multidrug-resistant *E. coli*. Compared to BCoV FREE herds: BCoV PSI herds had significantly higher odds of cough in all ages, and of diarrhea in young stock and cows, and a higher proportion of cows with fever. BCoV RI herds had significantly higher odds of diarrhea in young stock and cows and of cough in all ages. The results support previous research that freedom from BRSV and BCoV is beneficial for animal welfare and farm economy and possibly also mitigates AMR.

## 1. Introduction

The burden of disease in food animal production is higher for endemic than for epizootic diseases and they are the major targets for antimicrobial treatments [1]. The prevention and control of endemic diseases is therefore likely to be beneficial for productivity and animal welfare but also pivotal for counteracting antimicrobial resistance (AMR), since healthy animals do not need antimicrobial treatments.

Bovine respiratory syncytial virus (BRSV) and bovine coronavirus (BCoV) are two of the most important viral diseases of cattle worldwide [2,3]. The infections have serious effects on herd economy and animal welfare [4,5], and circulate annually in cattle herds in Sweden [6,7]. Infection with BRSV causes respiratory tract disease [3], while BCoV infection manifests with enteritis and variable degrees of respiratory tract involvement [2,5]. Infected animals are prone to secondary bacterial infections which may require antimicrobial treatments, that can select for resistant bacteria [3].

Researchers in Sweden and Norway have addressed the epidemiology of BRSV and BCoV, including prospects for control [8,9]. Both diseases are endemic in Swedish cattle herds and it has been suggested that very similar strains of BRSV and BCoV circulate in the cattle population [6,7] and that specific antibodies acquired remain detectable for years, even without reinfection [10,11]. All animals born in a herd after an infection has passed will be naïve and susceptible to a new infection. Herds free from infection are assumed to be hit harder on the introduction of the viruses due to a higher proportion of susceptible individuals in the herd, than herds that regularly encounter the virus, and thereby sustain a certain degree of herd immunity [5]. It is probable, however, that repeated re-infections in a herd affect the overall health and productivity, resulting in increased mortality and antimicrobial use. It might therefore be sounder, from an economical and an animal welfare perspective, to sustain a free status rather than having repeated infections to sustain immunity.

Voluntary control programs, based on biosecurity measures, have been suggested in Sweden and are already implemented in Norway [9,12]. To support the development of control programs for BRSV/BCoV, an evaluation of the benefits of such measures for herd health and productivity as well as for antimicrobial use and AMR is valuable. Previous studies have shown that BRSV and BCoV negatively affect productivity in dairy herds [5,13,14,15,16,17]. However, to the best of our knowledge, the effects on overall herd health, antimicrobial use, and AMR have not been evaluated. If preventive measures to sustain freedom from these viruses counteract AMR, this would be a strong incentive to implement control programs, not only for the dairy industry, but also for society. In addition, most research in this area has focused on the short-term effects of outbreaks and has not considered the effects of having repeated infections compared to being completely free. The aim of this study was to investigate the effects of recurring infections, recent infections, or freedom from BRSV or BCoV on herd productivity, health status, antimicrobial use and the occurrence of AMR. We hypothesize that freedom from these diseases would be beneficial regarding all these aspects.

## 2. Results

### 2.1. Herds and Dropouts

In all, 111 herds (17% of invited) agreed to participate and were enrolled in the study from October 2015–March 2016. The median herd size at enrolment was 65 cows. The herds were in the East (13%), South (39%) and North (48%) of Sweden [18]. Eight herds did not send in fecal samples at sampling 1 and were excluded from the study. An additional 16 herds dropped out at sampling 2, and another 11 at sampling 3, resulting in 76 herds that completed the study. The median herd size for these 76 herds was 58 cows and they were in the East (14%), South (41%) and North (45%) of Sweden. Specific reasons for dropouts were not sought. Up to date information about herd size could not be obtained from invited herds that did not participate and hence, a comparison between participating and non-participating herds was not possible.

### 2.2. Samples

The median number of days between milk samplings 1 and 2 was 367 (interquartile range (IQR): 348–392) and 367 (IQR: 351–382) between milk samplings 2 and 3. The median number of days between the last fecal sample in sampling 1 and the first in sampling 2 was 303 (IQR: 287–340) days, and between the last fecal sample in sampling 2 and the first in sampling 3 was 366 (IQR: 344–392) days. Of the 76 herds, 74, 74, and 70 submitted six fecal samples at samplings 1, 2 and 3, respectively. The remainder of herds submitted between two and five samples at each sampling. The sampled calves were between 0 and 100 days old with a median age of 14 (IQR: 7–23) days. 

### 2.3. Questionnaire Response Rate 

Of the 608 questionnaires sent to the 76 herds that completed the study, 595 were returned, resulting in a response rate of 98%. Of these, 77% were fully completed. The mean response rate for each question was 95%, ranging between 87% and 98% for different questions. The questions on incidence of hoof or leg disorders and dullness in young stock had more than 10% missing values and were excluded from further analyses. Questionnaire data are presented in Appendix A.

### 2.4. Herd Infection Status

In the first sampling of milk from three homebred primiparous cows (PPM), 30% of the 76 herds were both BRSV and BCoV antibody-positive, 32% were positive to BCoV only, 9% were positive to BRSV only, and 29% were negative to both viruses. For samplings 2 and 3, the corresponding results were 38%, 32%, 5%, and 25%, and 32%, 33%, 10%, and 25%, respectively. Based on the BRSV and BCoV antibody status in PPM, herds were categorized as free of infection (FREE), recently infected (RI) and past steadily infected (PSI) after the final sampling. In ten herds (13%), all three samples were antibody-negative to both viruses, and an additional nine herds (12%) were free from any of the virus infections during the study period. The number of herds in each category for each virus is presented in Table 1.

### 2.5. Antimicrobial Resistance

Quinolone-resistant *E. coli* (QREC) were isolated from calves on all but one farm. The mean proportion of calves within a farm that carried QREC was 49%, ranging from 0 to 100%. The median within-sample prevalence of QREC for individual calves was 0% (IQR: 0–0.003%) and ranged from 0 to 100%. 

Tetracycline-resistant *E. coli* (TREC) were isolated from calves on all but one farm. The mean proportion of calves within a farm that carried TREC was 81%, ranging from 0–100%. The median within-sample prevalence of TREC for individual calves was 0.09% (IQR: 0.0002–1.6%) but ranged from 0 to 100%. 

Of the 1331 randomly selected *E. coli* isolates subjected to antimicrobial susceptibility testing, 637 (48%) were resistant to one or more antimicrobials. The proportion of resistance, and the minimum inhibitory concentration reached by 50% of isolates (MIC50) and MIC90 are presented in Appendix A. Twenty-one percent (280 isolates) were multidrug-resistant (MDR).

### 2.6. Effect Associated with Infection Status

For outcomes where a statistical difference (*p* < 0.05) or a statistical trend (*p* < 0.1) was observed between FREE and PSI or RI herds, the direction of the effect of BRSV and BCoV infection status is shown in Table 2. Adjusted values, predicted probabilities, and hazard ratios (HR) and odds ratios (OR) are presented for each affected outcome in the text below.

#### 2.6.1. BRSV

Compared to FREE herds, PSI herds had significantly higher odds of cough in young stock (OR 5.8, *p* = 0.005), a higher proportion of QREC (4.9% vs. 1.2%, *p* = 0.025), but a lower proportion of cows with non-specific fever (0.079% vs. 0.41%, *p* = 0.002). Additionally, PSI herds tended to have lower milk yield (29.7 kg ECM/day vs. 31.9 kg ECM/cow and day, *p* = 0.087) than FREE herds and higher odds of diarrhea in calves (OR 1.5, *p* = 0.086) and young stock (OR 2.3, *p* = 0.095). Compared to FREE herds, RI herds had significantly higher odds of diarrhea in calves (OR 1.7, *p* = 0.013) and young stock (OR 2.6, *p* = 0.037), a higher proportion of QREC (5.0% vs. 1.2%, *p* = 0.012), higher odds of multidrug-resistant *E. coli* (OR 1.8, *p* = 0.038) and tended to have higher calf mortality in calves younger than two months (HR 1.4, *p* = 0.062) and in calves 2–6 months (HR 1.6, *p* = 0.075).

#### 2.6.2. BCoV

Compared to FREE herds, PSI herds had significantly higher odds of cough in calves (OR 2.8, *p* = 0.049), young stock (OR 6.3, *p* = 0.001), and cows (OR 13.1, *p* = 0.001), and of diarrhea in young stock (OR 2.8, *p* = 0.036) and cows (OR 3.8, *p* = 0.005), and a higher proportion of cows with non-specific fever (0.48% vs. 0.064%, *p* = 0.001). Additionally, PSI herds tended to have a higher proportion of cows with hoof or leg disorders (2.8% vs. 2.1%, *p* = 0.093), and of antimicrobial-treated cows (3.3% vs. 2.4%, *p* = 0.080), but a lower mortality for calves younger than two months (HR 0.7, *p* = 0.094). Compared to FREE herds, RI herds had significantly higher odds of diarrhea in young stock (OR 4.7, *p* = 0.004) and cows (OR 3.8, *p* = 0.011) and of cough in calves (OR 7.1, *p* = 0.002), young stock (OR 7.2, *p* = 0.003), and cows (OR 11.4, *p* = 0.009). 

## 3. Discussion

In general, the study supported our hypothesis that freedom from BRSV and BCoV in dairy herds is associated with better health status, higher productivity, fewer antimicrobial treatments, and a lower occurrence of AMR. Of the 28 outcomes evaluated, five supported the hypothesis for BRSV with statistical significance and six for BCoV, respectively. In addition, three statistical trends supported the hypothesis for BRSV and two for BCoV, whereas only one outcome variable with statistical significance refuted the hypothesis for BRSV and one trend for BCoV. To the best of our knowledge, this is the first study that simultaneously evaluated associations between the presence of specific infections and a broad range of outcome variables, including AMR. 

### 3.1. Effects on Productivity

In the present study, there was a statistical trend towards lower milk production in BRSV PSI herds but no difference between FREE and RI herds. For BCoV, we found no effect on milk production. The impact of BRSV and BCoV infections on milk production in dairy herds has been evaluated in earlier studies, some of them showing transient or subtle decreases in milk yield [13,16,17,19,20]. Although the proposed effect of BRSV on milk production in this study is only a trend, it confirms the previous research results and suggests that the effects on milk yield are subtle or transient and that either daily milk yield data or a larger sample size may be required to find significant associations.

Counter to our hypothesis, neither we nor Ohlson et al. [15] observed an effect of BRSV or BCoV on fertility. To use fertility measures as outcome variables is challenging because they often are blurred by the preferences, skills and routines of each farmer and may not truly reflect any effect of a virus infection.

### 3.2. Effects on Morbidity

Both BRSV and BCoV infections affected, or tended to affect, several morbidity outcomes. BRSV RI herds experienced diarrhea in calves and young stock significantly more often than FREE herds. Diarrhea has been reported previously [21] but is normally not a symptom of BRSV infection. Possibly, diarrhea was not a direct consequence of BRSV infection, but due to an overall lower level of biosecurity, hygiene, and management in RI than in FREE herds. There was also a tendency for higher calf mortality in RI herds, which confirms previous research [13,22]. Cough is a common symptom of BRSV, and although there was a tendency for a higher incidence in young stock in PSI herds, there were no associations for the other animal categories in PSI herds and not for any category in RI herds. Although cough is just one of many symptoms of BRSV, this result is surprising. There are, however, several other causes for coughing in dairy herds, and the single effect of BRSV might be difficult to identify. In BRSV PSI herds, we observed a significantly lower occurrence of non-specific fever in cows compared to FREE herds. This is contrary to our hypothesis, but it is possible that FREE herds are more wary of disease events, whereas in PSI, some cases of fever may be missed out due to a lower degree of awareness. 

Diarrhea, cough, and fever are common symptoms of BCoV [2,5] and it was not surprising that these symptoms were significantly more common in BCoV PSI and RI herds than in FREE herds. Interestingly, there was no difference in the occurrence of diarrhea in PSI and RI herds, although less diarrhea could be expected in PSI herds due to an assumed degree of herd immunity. This indicates that fully protective immunity of individual animals is short-lived, and previously infected cows are susceptible to BCoV, which agrees with previous findings [19]. Diarrhea was manifested predominantly in older animals, and not in calves, which also is in accordance with previous findings [5]. Contrary to our hypothesis, BCoV PSI herds tended to have lower death hazard in calves under the age of two months, which partly agrees with Beaudeau et al. [13] who found a lower death hazard in calves and young stock in RI herds. As discussed by Beaudeau et al. [13], there may be management factors or herd characteristics unrelated to infection status that can explain the lower mortality in infected herds.

In this study, no association was found with udder disease and milk somatic cell count (SCC), feed-related disorders in cows, metritis/retained fetal membranes, abortions, other disease in calves, young stock mortality, and culling of cows. As the sample size was small, missed associations cannot be ruled out.

### 3.3. Effects on Antimicrobial Use and AMR

Although both BRSV and BCoV infection status were associated with higher odds of cough and diarrhea, infection status was overall not associated with a higher antimicrobial treatment incidence. An exception to this is a tendency for BCoV to be associated with the increased treatment of cows in PSI herds, which is in accordance with previous findings that BCoV infections led to a higher treatment incidence in feedlot cattle [23] and beef calves [24].

Although we did not observe a higher incidence of antimicrobial treatment in BRSV-positive herds, MDR *E. coli* were more common in BRSV RI herds than in FREE herds and the proportion of QREC in the fecal flora of calves was higher in both BRSV RI and PSI herds. Antimicrobial use is considered a strong driver for resistance, but resistant bacteria can also spread and persist on farms where antimicrobial use is low and can be associated with poor farm biosecurity and hygiene [25]. This agrees with a previous study from our group showing that poor external biosecurity and poor hygiene was associated with a higher occurrence of QREC on dairy farms [26]. The lower occurrence of antimicrobial resistance in BRSV FREE herds could therefore be due to an assumed higher level of biosecurity in FREE compared to PSI herds.

### 3.4. General Discussion

The present study adds to, and supports previous research indicating that efforts to reduce the occurrence of BRSV and BCoV in dairy herds would improve animal health and production, reduce antimicrobial treatment rates, and possibly also mitigate the spread of antimicrobial resistance. It is rather clear that, for most outcome variables, dairy herds would do better sustaining a FREE status. The results also indicate that having repeated infections (PSI) is not better than going from a free (FREE) to an infected status (RI), but may instead be worse in the longer term. Available scientific evidence therefore supports efforts to establish control programs for BRSV and BCoV in dairy herds. Voluntary control programs, based on biosecurity measures, have been suggested in Sweden and are already implemented in Norway. The feasibility of control is demonstrated by the fact that 58% of the herds in the study upheld a BRSV FREE status throughout the two-year study period, 36% a BCoV FREE status and 25% a FREE status for both diseases. Of the 76 herds that completed the study, 10 herds entered in a completely free state and sustained it throughout the study. This clearly shows that it is possible to remain free for longer periods, which is also supported by a previous study [9].

Observational studies, as the present, are not ideal for investigating causal relationships because it may be difficult to know the direction of causality and it could also be that virus infection is a proxy for unknown factors. Thus, an infected herd may have lower productivity and impaired health status due to poorer management than FREE herds; for example, gaps in biosecurity increasing the risk of introducing infections and antimicrobial-resistant bacteria. Observational studies, on the other hand, are more likely to represent the true field situation than experimental studies. However, the conclusions of this study are only indicative and should be used in conjunction with previous and future research.

As morbidity and treatment data from SOMRS are incomplete [27], we decided to rely on farmers’ reports for these data. However, a limitation of the study was that we did not clearly define morbidity outcomes. Farmers are obliged to keep treatment journals and should be able to report accurate data but may have had different thresholds for reporting morbidity, which could have biased the results of the study.

As the outcome indicator of AMR, we used the antimicrobial susceptibility of enteric *E. coli* from calves. This indicator is commonly used to monitor AMR in farm animals and considered to reflect the selection pressure from the use of antimicrobials in animal populations [28]. Although this indicator does not reflect the complete resistome of the animals, it is a simple and well-established way to evaluate the occurrence of AMR in farm animals. It should also be noted that age is a well-known risk factor for the occurrence of antimicrobial-resistant bacteria in feces from calves [26,29], and to reduce variation due to age, we aimed at including calves of a similar age (below 1 month). However, we also received samples from older calves, and to avoid the loss of information, these samples were included in the analyses. To account for the variation, age was included as a covariate in all models where it significantly affected the outcome.

By using a significance level of 10% instead of the more common 5%, we deliberately increased the risk of a type I error (overestimation) while decreasing the risk of type II errors (underestimation). Additionally, by testing many outcomes simultaneously, the risk of false associations increases. We believe, however, that falsely declaring no effect of virus infection is worse than reporting statistical trends that may be overestimated. As mentioned earlier, conclusions of this study should not stand alone but rather be used in parallel with conclusions from previous and future research in the same field.

## 4. Materials and Methods 

### 4.1. Study Design

The study was performed as an ambi-directional cohort study. The virus exposure was assessed using repeated testing for BRSV and BCoV antibodies in PPM during a period of two years. Each herd submitted milk samples three times: 1st at enrollment (October 2015–March 2016); 2nd about one year later (November 2016–June 2017); 3rd about two years after the first (November 2017–June 2018). 

Herds were categorized for the putative presence of viral infections during this period based on antibody status to BRSV and BCoV in PPM at the three samplings (Table 1). Herds were considered RI if antibody-negative at any of sampling 1 or 2 and positive in sampling 3 and PSI if antibody-positive in all three samplings. RI and PSI herds had at least one viral infection during the period, but PSI herds were presumed to enter the study with more immune animals than RI herds. Herds were considered FREE if antibody-negative in sampling 3, regardless of the result in samplings 1 and 2. By assessing PPM, the presence of antibodies suggests that the herd has had at least one viral infection during the two preceding years, assuming first calving at the age of two years. Thus, a negative third sampling indicates that no virus infection has occurred during the period.

Meanwhile, data on production, health, antimicrobial use as well as fecal samples to determine the presence of antimicrobial-resistant *E. coli* were collected and used to retrospectively assess the effects of the BRSV/BCoV exposure levels.

The experimental design and handling of animals involved no invasive procedures, and an ethical approval was therefore not required.

### 4.2. Selection of Herds

In October 2015, 635 Swedish dairy herds were invited by postal service to participate in the study. Only herds with at least 30 lactating cows and affiliated to the Swedish official milk recording system (SOMRS) were invited. To maximize the variation in antibody status to BRSV and BCoV of the herds, they were selected for invitation based on antibody status in the most recent nationwide screening of bulk tank milk which was performed in 2013 (A. Ohlson, unpublished data). Although some herds might have changed status since 2013, many herds remain antibody-negative for several years [30] and this approach increased the likelihood of recruiting negative farms to the study. First, all herds that were considered free from both infections and all BRSV-positive but BCoV-negative herds were selected for invitation (25% of all invited herds). Thereafter, the same number of BCoV-positive but BRSV-negative herds was selected (25%) and finally, the same number of herds positive to both infections (50%). In the latter two groups, herds were selected randomly from all eligible herds. The invitation included sampling material for the first milk sampling. Invited herds that wanted to participate did so by sending in that first milk sample, resulting in a convenience sample of herds.

We aimed to include the number of herds needed to be able to show an effect of BRSV/BCoV status on the presence of antimicrobial-resistant bacteria. The prevalence of tetracycline resistance among random *E. coli* isolates from healthy calves aged 7–28 days has been shown to be 23% [31] with the intra-herd correlation of 0.24 (unpublished results). Assuming then that tetracycline resistance was 30% in infected herds and 15% in free herds, we estimated that it was necessary to sample at least 10 calves per herd in 40 infected and 40 free herds with a power of 80% and confidence level of 95%.

### 4.3. Sampling and Laboratory Analyses

Sampling materials, instructions and submission forms were provided by the National Veterinary Institute (SVA) in Uppsala, Sweden. Samples were collected by farmers/farm staff and sent at ambient temperature by postal services to SVA. Samples were to be sent on Sundays to Thursdays, to ensure delivery at the laboratory before the next weekend.

Milk samples: The levels of BCoV- and BRSV-specific antibodies were assessed using pooled PPM from three homebred primiparous cows per herd and sampling, a method previously validated for this purpose [28]. Ten-milliliter test tubes containing 10 microliters of the preservative agent Bronopol (2-bromo-2-nitropropane-1.3-diol, SVA, Uppsala, Sweden) were used for sampling. The tubes were marked with a herd-specific code. 

On arrival at the laboratory, the samples were stored at −20 °C until analysis. All samples were analyzed for immunoglobulin G antibodies to BRSV and BCoV by commercially available indirect ELISAs (Svanovir BRSV-AB (Batch A32465, A67520) and Svanovir BCoV-AB (Batch A26896, A65331), Boehringer Ingelheim Svanova, Uppsala), respectively. The sensitivity and specificity for the BRSV ELISA was 94% and 100% [32] and for the BCoV ELISA 84.6% and 100% [11], respectively. The optical density (OD) at 450 nm was corrected by the subtraction of the negative control antigen OD. To adjust for day-to-day variations, the percentage positivity (PP) was calculated as follows: (sample corrected OD/positive control corrected OD) × 100. For BRSV/BCoV, the cut-off was set to PP equal to 20. Herds were considered positive if the PP value was ≥20 and negative otherwise. The same cut-off has been used in previous studies to categorize herds as negative or positive for BRSV/BCoV infection [13,30].

Fecal samples: In each herd, at each of the three sampling occasions, the herd-level antimicrobial susceptibility of *Escherichia coli* (*E. coli*) was assessed using fecal samples from six preweaned calves (in total 18 samples per farm during the study period). The calves should be younger than 1 month of age, healthy, and never treated with antimicrobials. Fecal samples were collected from the rectum using Amie’s charcoal culture swabs (Copan Diagnostics Inc., Murrieta, CA, USA) and were submitted with information on the sampling date and birth date of each calf. 

Fecal samples were analyzed directly upon arrival at the laboratory. For each sample, the within-sample prevalence of *E. coli* resistant to quinolones (QREC) or tetracycline (TREC) was determined (see below) and in addition, the susceptibility to 13 antimicrobials for one *E. coli* from each sample was analysed (see below). 

To determine the within-sample prevalence of QREC and TREC, tenfold dilutions of each sample were plated on Petrifilm Select *E. coli* Count (SEC plate; 3M Microbiology Products, St. Paul, MN, USA) with and without nalidixic acid or tetracycline. The procedure identifies *E. coli* with MICs above the epidemiological cut-of value (ECOFF) for nalidixic acid (16 mg/L) or tetracycline (8 mg/L) established by the European Committee on Antimicrobial Susceptibility Testing (EUCAST—www.eucast.org (accessed on 10 September 2015)). The method was modified from Wu et al. [33] and was previously used for QREC in our laboratory [26]. In brief, swabs were vortex mixed in 3 mL of 0.9% saline to release fecal content. From this non-diluted suspension tenfold dilutions down to 10^−6^ were made in 0.9% saline. The total CFU count of *E. coli* was determined by plating 1 mL of dilutions 10^−4^ and 10^−6^ onto the bottom film of an SEC plate. 

The CFU of QREC was determined as follows. A stock solution of nalidixic acid sodium salt at 672 µg/mL (Sigma-Aldrich Co, St. Louis, MO, USA) was prepared. Fifty microliters stock solution was added to 1000 µL of the non-diluted and the 10^−2^ suspension to obtain a final concentration of 32 µg/mL nalidixic acid. The entire volume of this suspension (1050 µL) was plated onto the SEC plates as described above. The CFU of TREC was determined as follows. A stock solution of tetracycline hydrochloride at 1344 µg/mL (Sigma-Aldrich Co, St. Louis, MO, USA) was prepared. This solution was added to sample dilutions 10^−2^, 10^−4^, and 10^−6^ to obtain a final concentration of 64 µg/mL tetracycline and plated as described above. The concentrations of nalidixic acid and tetracycline were chosen to only allow the growth of *E. coli* with MICs above the ECOFFs for these antimicrobials. Hence, *E. coli* colonies on plates with nalidixic acid were considered QREC and colonies on plates with tetracycline were considered TREC. Plates were incubated for 18 to 24 h at 42 °C and dark green to light blue–green colonies were counted. The within-sample prevalence of QREC and TREC in a sample was obtained by dividing the CFU of *E. coli* on plates with antimicrobials by the CFU on plates without antimicrobials.

One colony of putative *E. coli* was selected from a SEC plate without antimicrobial and after subculture susceptibility tested by microdilution according to the recommendations of the Clinical and Laboratory Standards Institute [34] using VetMIC panels (National Veterinary Institute, Uppsala, Sweden) and cation-adjusted Mueller Hinton broth (Becton Dickinson, Cockeysville, MD, USA). Antimicrobials and ranges are given in Appendix A. Quality control, using the reference strain *E. coli* ATCC 25922, was conducted in parallel with each batch of isolates tested; all results were within acceptable ranges. ECOFFs (Appendix A) were used to classify isolates as wild-type and non-wild-type, where the latter were considered resistant. An isolate was defined as MDR if resistant to three or more antimicrobial classes.

### 4.4. Data Collection and Preparation

For each herd, data on individual milk, fat, and protein yield, SCC, and reproductive events (calvings, breedings, and pregnancy checks), entrance and exit dates as well as reason for exit for all animals present at any time during the study period were obtained from SOMRS. Data used were from the period between the first and the third milk sampling, irrespective of individual birth dates, reproductive cycles, or lactation stages. Reproductive failure and mortality data were defined and prepared according to Beaudeau et al. [13]. The level of energy-corrected milk (ECM) was calculated for each cow using the following formula:kg ECM = kg milk × 0.25 + fat% × 12.2 + protein% × 7.7

Data on morbidity and antimicrobial treatment were obtained from the farms via repeated questionnaires. At eight occasions during the 2-year study period, from November 2015 to October 2017, farmers/farm staff responded to questionnaires on the health status of the herd. Each questionnaire covered the preceding two months, but for practical reasons, the summer months (May–August) were not covered. Thus, information from each herd was gathered for 16 months of the 24-month study period. BRSV and BCoV infections are less common in the summer months and this period was excluded to ease the workload for the farmers/farm staff. The questionnaire contained questions on the incidence of diarrhea and cough, in calves under the age of six months, in young stock from six months to calving/slaughter (including bulls if present), and cows (lactating and dry). The incidence was rated as “none”, “a few”, “one fourth of the age group”, “half of the age group”, and “more than half of the age group”. For statistical analysis, these questions were transformed to binary variables, such as no diarrhea/cough versus diarrhea/cough in a few or more animals. 

The questionnaire also contained questions on the number of cows that had experienced the following disease events: mastitis/high SCC, hoof or leg disorders, feed-related disorders (ketosis or inappetence), metritis/retained fetal membranes, fever without obvious reason, and abortion (gestation length < 260 days). Additionally, questions on the number of young stock with hoof or leg disorders or with dullness without obvious reason were included, and on the number of calves up to six months with hoof or leg disorders, umbilical infections, or dullness without obvious reason. Finally, questions regarding the number of lactating cows, dry cows, young stock and calves that had been treated with antimicrobials, irrespective of type, during the period of concern was included. The questions regarding dullness, hoof or leg disorders and umbilical infections in calves had few non-zero variables and were therefore merged into one outcome variable—other diseases in calves. In May each year, a question on vaccination against BRSV or BCoV in the preceding year was added because vaccination could interfere with the result of the antibody testing. After the first postal questionnaire, farmers/farm staff could choose to receive and respond to subsequent questionnaires by post, via the web-based platform Easyresearch (Questback AB, Stockholm, Sweden), or by telephone. The number of disease events and antimicrobial treatments were adjusted to the mean number of animals in the specific category present in the herd.

### 4.5. Statistical Analyses

The effects of BRSV and BCoV were evaluated separately, but it was assumed that all outcomes may be affected by both infections. Therefore, both BRSV and BCoV statuses were forced into all models and the estimates of the effect of one infection were adjusted for the possible effect of the other. Due to the time of infection, all observations within the herd-specific study period were considered as potentially affected by virus infection. Three herds vaccinated against BCoV and one against BRSV and were included as a separate group (results of statistical modelling not presented). 

Different statistical models were used for different outcomes and each outcome was adjusted for the potential effects of other important covariates based on earlier studies or on biological assumptions, as shown in Table 3. All models were reduced by stepwise backward elimination to only include covariates that significantly affected the outcome or covariates that changed the estimates of remaining covariates by more than 30% (confounding). Due to the small sample size, no interactions were considered [35].

Where necessary, to meet the assumptions of normally distributed residuals, continuous variables were transformed (Table 3). Depending on the model type, either hierarchical mixed models considering the herd and individual as random effects, or a robust variance estimator was used to account for clustering of observations at the herd or individual level. For each outcome analyzed, statistical model type, epidemiological unit, and explanatory or confounding variables included in the model are described in Table 3. All analyses were performed in Stata 15 (StataCorp, Midtown, TX, USA. 2017. Stata Statistical Software: Release 15. College Station, TX: StataCorp LLC). The effect on milk production, age at first calving, somatic cell count, and calving interval was assessed by mixed effects linear regression using the MIXED command. A compound symmetry covariance matrix was used to account for correlations between repeated measures from the same individual. The effect on diarrhea and cough in different age categories and multidrug resistance was assessed by mixed logistic regression using MEGLM. Cox proportional hazard models using STCOX were used to assess the effect on mortality and culling. Finally, fractional probit regression models using FRACGLM were conducted to assess the effect on all the other disease outcomes and on antimicrobial treatments. A significance level of 5% was used to define a statistically significant difference and a level between 5 and 10% was used to define a statistical trend/tendency.

## 5. Conclusions

Sustaining a free status for BRSV and BCoV is beneficial for the production and health of dairy farms, and is possibly also a way to reduce the need for antimicrobial treatments. Measures to uphold a free status likely also reduce the spread of other infectious diseases and antimicrobial-resistant bacteria. In a wider perspective, improved animal health in cattle would reduce the incidence of antimicrobial treatments and thereby the selection pressure towards resistant bacteria.

## Figures and Tables

**Table 1 antibiotics-10-00641-t001:** Categorization of herd infection status according to the herd antibody status in each milk sampling for bovine respiratory syncytial virus (BRSV) and bovine coronavirus (BCoV) in 76 dairy herds.

Milk Sampling Occasion Antibody-Positive (+) or Negative (−)			Herd Status ^1^	BRSV (Number of Herds)	BCoV (Number of Herds)
1	2	3			
−	−	−	FREE	29	14
+	−	−	FREE	8	4
+	+	−	FREE	3	9
−	+	−	FREE	4	0
−	−	+	RI	2	5
+	−	+	RI	4	0
−	+	+	RI	11	10
+	+	+	PSI	14	31
			VACCINATED ^2^	1	3

^1^ FREE = free of infection; RI = recently infected during the period; PSI = past steadily infected during the period. ^2^ Vaccinations with vaccines against either of bovine coronavirus or bovine respiratory syncytial virus.

**Table 2 antibiotics-10-00641-t002:** A schematic overview of the effects of infection status (PSI or RI) for bovine respiratory syncytial virus (BRSV) or bovine coronavirus (BCoV) infection on different outcomes related to herds with a FREE status. Upwards arrow denotes a higher value than in herds with a FREE status and downwards a lower value. Two arrows denote a statistically significant effect (*p* < 0.05), one arrow a statistical trend (0.05 < *p* < 0.1), and a dash neither a statistically significant effect nor a statistical trend (*p* > 0.1).

Outcome Variable	BRSV		BCoV	
	PSI	RI	PSI	RI
**Productivity**				
Milk production	↓	-	-	-
Reproductive failure	-	-	-	-
Calving interval	-	-	-	-
Age at first calving	-	-	-	-
**Health Status**				
Diarrhea calves	↑	↑↑	-	-
Diarrhea young stock	↑	↑↑	↑↑	↑↑
Diarrhea cows	-	-	↑↑	↑↑
Cough calves	-	-	↑↑	↑↑
Cough young stock	↑↑	-	↑↑	↑↑
Cough cows	-	-	↑↑	↑↑
Udder disease	-	-	-	-
Somatic cell count	-	-	-	-
Non-specific fever cows	↓↓	-	↑↑	-
Hoof or leg disorders cows	-	-	↑	-
Feed-related disorders cows	-	-	-	-
Metritis/retained fetal membranes	-	-	-	-
Abortions	-	-	-	-
Other disease ^1^ calves	-	-	-	-
Mortality calves 1–59 days	-	↑	↓	-
Mortality calves 60–179 days	-	↑	-	-
Mortality young stock 180–455 days	-	-	-	-
Culling cows	-	-	-	-
**Antimicrobial Treatments**				
Cows	-	-	↑	-
Calves	-	-	-	-
Dry cows	-	-	-	-
**Antimicrobial Resistance**				
Within-sample prevalence of QREC ^2^	↑↑	↑↑	-	-
Within-sample prevalence of TREC ^3^	-	-	-	-
Multidrug resistance	-	↑↑	-	-

^1^ Hoof or leg disorders, umbilical infections and dullness without obvious reason. ^2^ Quinolone-resistant *Escherichia coli.*
^3^ Tetracycline-resistant *Escherichia coli.*

**Table 3 antibiotics-10-00641-t003:** Overview of statistical models, epidemiological units and included explanatory or confounding variables for each dependent variable.

Dependent Variable (Transformation if Necessary)	Statistical Model	Epidemiological Unit	Explanatory Variables ^1^
**Productivity**			
Milk production	Linear	Cow	a, b. c, d, e, f, h, i *, j *
Reproductive failure	Logistic	Cow	a, b, c, e, i *, j *
Age at first calving (inverse)	Linear	Cow	a, b, f, g *, i *, j *
Calving interval (inverse cubic)	Linear	Cow	a, b, e, f, g *, i *, j *
**Health Status**			
Diarrhea calves	Logistic	Herd	a, b, g *, i, j
Diarrhea young stock	Logistic	Herd	a, b
Diarrhea cows	Logistic	Herd	a, b, g *, i, k
Cough calves	Logistic	Herd	a, b, g *, i, j*
Cough young stock	Logistic	Herd	a, b, g *
Cough cows	Logistic	Herd	a, b, e*, g *, j *
Udder disease	Fractional probit	Herd	a, b, g, i, j *
Somatic cell count (log)	Linear	Cow	a, b. c, d, e, f, g, i, j *
Non-specific fever—cows	Fractional probit	Herd	a, b, e, i, j, k
Hoof and leg disorders—cows	Fractional probit	Herd	a, b, g *, i *, j *, k
Feed-related disorders—cows	Fractional probit	Herd	a, b, e *, g *, i, j *, k
Metritis	Fractional probit	Herd	a, b, e *, g *, i, j *
Abortions	Fractional probit	Herd	a, b, e *, i *, j *, k
Other disease ^2^—calves	Fractional probit	Herd	a, b, e *, g * j *, k
Mortality calves 1–59 days	Cox proportional hazards	Calf	a, b, i, j
Mortality calves 60–179 days	Cox proportional hazards	Calf	a, b, g *, i, j
Mortality young stock 180–455 days	Cox proportional hazards	Young stock	a, b, j *, l
Culling cows	Cox proportional hazards	Cow	a, b, g *
**Antimicrobial Treatments**			
Cows	Fractional probit	Herd	a, b, e *, g, i, j *
Calves	Fractional probit	Herd	a, b, g, i *, j
Dry cows	Fractional probit	Herd	a, b, e*, g *, i, j *
**Antimicrobial Resistance**			
Within-sample prevalence QREC ^3^	Fractional probit	Calf	a, b, e *, f, g *, i *, j, l, m *, n, o *
Within-sample prevalence TREC ^4^	Fractional probit	Calf	a, b, f *, g *, i, m *, o *
Multidrug resistance	Logistic	Calf	a, b, i, j *, n, o, p

^1^ a = herd antibody status to bovine respiratory syncytial virus, b = herd antibody status to bovine coronavirus, c = parity, d = days in milk, e = breed, f = season, g = milk production, h = somatic cell count, i = herd size, j = geographic location, k = period, l = gender, m = calf age, n = calf use of antimicrobials, o = cow use of antimicrobials, p = sampling occasion, * Included as confounder. ^2^ Hoof or leg disorders, umbilical infections and dullness without obvious reason. ^3^ Quinolone-resistant *Escherichia coli.*
^4^ Tetracycline-resistant *Escherichia coli*.

## Data Availability

The data presented in this study are available on request from the corresponding author.

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
