# Peer review of "Associations between Bovine Coronavirus and Bovine Respiratory Syncytial Virus Infections and Productivity, Health Status and Occurrence of Antimicrobial Resistance in Swedish Dairy Herds"

_antibiotics, 2021, doi:10.3390/antibiotics10060641_

Round 1
Reviewer 1 Report
This is an interesting survey of a selection of Swedish Dairy farms in relation to their BRSV and BCoV status. The study is difficult to read, since the M&M are presented after the results. Thus, when you read the M&M to get an understanding of study design, then you have to search in results to find explanation for acronyms. Furthermore in the M&M in the Study design section you have to go to sampling to understand study design. I assume that this is the order required by journal, but maybe just thinking about how a reader will understand text without forcing the reader to hunt around in text can help the reading of the article.
You have evaluated correlations between BRSV and BCoV status and various outcomes, but there may be confounding. It may be that a lower risk of BRSV and BCoV is found due to overall biosecurity measures and that these biosecurity measures also lowers the risk for other diseases. The it is not per see the BRSV/BCoV status that leads to higher morbidity/mortality/AMR, but maybe other diseases as well. This should be clearly pointed out. You state that you now have evidence to establish Control programmes for BRSV and BCoV, but this is not clear evidence for the reader. Maybe you have more evidence for a higher level of biosecurity. Unfortunately there are no information regarding biosecurity levels in the farms. This should be discussed. You have discussed some of the confounding in the discussion, but then in the conclusion you make a statement is that is not supported by this study, due to confounding. I would also recommend that you discuss some about the study participation by herds, and the impact of drop-outs and non-participation. Furthermore, it is unclear to the reader how many herds are using Rotavec vaccine and how this was included in the statistical models.
Author Response
Response to Reviewer 1
We thank the reviewer for constructive criticism and useful comments. Please see our responses below.
Comments and Suggestions for Authors
- This is an interesting survey of a selection of Swedish Dairy farms in relation to their BRSV and BCoV status. The study is difficult to read, since the M&M are presented after the results. Thus, when you read the M&M to get an understanding of study design, then you have to search in results to find explanation for acronyms. Furthermore, in the M&M in the Study design section you have to go to sampling to understand study design. I assume that this is the order required by journal, but maybe just thinking about how a reader will understand text without forcing the reader to hunt around in text can help the reading of the article.
Response: We agree with the reviewer that it would have been better to place the M&M section before “Results” as is conventional, but we must follow the instructions of the journal and place it at the end of the paper.
- You have evaluated correlations between BRSV and BCoV status and various outcomes, but there may be confounding. It may be that a lower risk of BRSV and BCoV is found due to overall biosecurity measures and that these biosecurity measures also lowers the risk for other diseases. The it is not per see the BRSV/BCoV status that leads to higher morbidity/mortality/AMR, but maybe other diseases as well. This should be clearly pointed out.
Response: We agree that it is not only freedom from infections due to BRSV or BCoV that will lead to an overall lower burden of disease in free herds. The assumed higher level of biosecurity needed to uphold a free status likely also mitigates introduction of other diseases and possibly also bacteria resistant to antimicrobials. We think that this is already quite clearly pointed out on lines 257-264 (“Observational studies, as the present, are not ideal for investigating causal relationships because it may be difficult to know the direction of causality and it could also be that virus infection is a proxy for unknown factors. Thus, an infected herd may have lower productivity and impaired health status due to poorer management than FREE herds, for example gaps in biosecurity increasing the risk of introducing infections and antimicrobial resistant bacteria.”) and on lines 239- 241 (“The lower occurrence of antimicrobial resistance in BRSV FREE herds could therefore be due to an assumed higher level of biosecurity in FREE compared to PSI herds.”).
However, we have tried to clarify this in Conclusion. See comment below.
- You state that you now have evidence to establish Control programmes for BRSV and BCoV, but this is not clear evidence for the reader.
Response: In the Introduction it is stated: “Researchers in Sweden and Norway have addressed the epidemiology of BRSV and BCoV, including prospects for control [8, 9]” (lines 54-55). And a couple of lines lower: “Voluntary control programs, based on biosecurity measures, have been suggested in Sweden and are already implemented in Norway [9, 12]”. We think that these three references provide the scientific basis for the feasibility of establishing control programs for BRSV and BCoV. The present study was intended to provide data concerning the benefits of such programs but in our opinion we think that a detailed account of the specific details presented in these three references would be out of scope of the present paper.
- Maybe you have more evidence for a higher level of biosecurity. Unfortunately there are no information regarding biosecurity levels in the farms. This should be discussed.
Response: We did not collect data on biosecurity levels in the herds. Although it can be assumed that the biosecurity level is higher in FREE herds we can therefore not discuss this in more detail.
- You have discussed some of the confounding in the discussion, but then in the conclusion you make a statement is that is not supported by this study, due to confounding.
Response: To make this clearer we have rephrased the Conclusion to: “Sustaining a free status for BRSV and BCoV is beneficial for production and health of dairy farms, and possibly also a way to reduce the need for antimicrobial treatments. Measures to uphold a free status likely also reduce the spread of other infectious diseases and antimicrobial resistant bacteria. In a wider perspective, improved animal health in cattle would reduce the incidence of antimicrobial treatments and thereby the selection pressure towards resistant bacteria”.
- I would also recommend that you discuss some about the study participation by herds, and the impact of drop-outs and non-participation.
Response: As stated on lines 85-88 we did not collect information on drop-outs or on herds not wanting to participate in the study. Unfortunately, any discussion on this would therefore be highly speculative.
- Furthermore, it is unclear to the reader how many herds are using Rotavec vaccine and how this was included in the statistical models.
Response: In Table 1 it is shown that 1 herd vaccinated against BRSV and 3 herds against BCoV. This information is also given and further discussed on lines 447-448.
Reviewer 2 Report
This article is entitled : “Associations between Bovine coronavirus and Bovine respiratory syncytial virus infections and productivity, health status and occurrence of antimicrobial resistance in Swedish dairy herds”. Its aim was “to investigate the effects of recurring infections, recent infections, or freedom from BRSV or BCoV on herd productivity, health status, antimicrobial use and occurrence of AMR.”
As general comments :
1/ Although interesting this paper lacks clarity. Several points in the material and methods needs clarification and results should be better presented in order improve the reader’s understanding.
2/ The line number are missing, which make the review difficult.
3/ You should have your text read by a native speaker. As an example, I believe the sentence “freedom from BRSV” is not correct in English.
Please find below more specific comments :
Abstract:
“BRSV RI herds had significantly higher odds of diarrhea in calves and young stock” : You should clarify compared to what? (this is also a comment latter in your results).
Introduction:
You should add information on the dairy industry in Sweden (economical importance, antimicrobial use regulations (specifically beta-lactams and fluoroquinolones which are classifies as important in human medicine), etc…)
“All animals born in a herd after an infection has passed will be naïve and susceptible to a new infection”: You should clarify this sentence; it seems in contradiction with the sentence just before.
“that specific antibodies acquired remain detectable for years, even without reinfection” this sentence seems in contradiction with what you say latter in the discussion about short lived immunity. Please clarify. Are the immunity mechanisms the same for both virus, etc…?
Materiel and Methods:
If I am not mistaken, your design is a cohort prospective and not retrospective (you assessed the herds status during the study).
“they were selected for invitation based on antibody status” could you clarify which status ?
I am not sure why you wanted to “increased the likelihood of recruiting negative farms to the study”.
Did you consider the sensitivity and susceptibility to your test at the herd level? Indeed, in a large herd where you test for three PPM, you will have less chances of detecting your outcome than in a small herd where you test 3 PPM. (you could at minimum mention it in the discussion).
It is not clear how you establish the number of herds (40 per group) and why in this part of the materiel and method you did not separate RI and PSI whereas you did in the other part of the study. It is also unclear why you chose the prevalence of tetracycline resistance to measure the number of calves you need by herd.
Why did you not use also an ESBL/AmpC enrichment? Could you add some information on antimicrobial use in Sweden? Maybe it would help understanding your choice.
Table 2 : P < 0.5 (did you mean 0.05?). Same remark for the other P.
Could you provide a reference to justify that the small size sample is a good reason not to consider interaction?
Results:
In the paragraph: 2.4:
For the sampling 3: 32%, 33%, 11%, and 25% makes 101%.
In paragraph 2.5 :
“Quinolone-resistant E. coli (QREC) were isolated from calves on all but one farm”. This is unusual and should be better explain with antimicrobial use in this country
In the paragraph, 2.6.1,
It is not clear to me if you also tested risk factors for the herds that were positive for both BCoV and BRSV as an exposure, if you did not could you tell why.
when you say : “Recently infected herds had significantly higher odds of diarrhea in calves” could you say significantly higher odds than what is it FREE and PSI together or just one of them and if so which one?
The same remark could be done for the paragraph 2.6.2.
I do not understand what your final model is. Maybe you should use a table to present these results to clarify.
I believe you did not present the results of the CFU, why did you perform it?
Discussion :
“but due to an overall lower level of biosecurity, hygiene, and management in RI than FREE herds.” It seems to me that FREE herds are supposed to have a higher level of biosecurity. Could you clarify this sentence? And indeed you say so later in you manuscript : “The lower occurrence of antimicrobial resistance in BRSV FREE herds could therefore be due to an assumed higher level of biosecurity in FREE compared to PSI herds.”
“and possibly also mitigate spread of antimicrobial resistance” : you have no data to support this sentence.
“therefore supports efforts to establish control programs for BRSV and BCoV in dairy herds” : What do you propose as control programs? Is vaccination a part of it? Although I understand you do not have many data on vaccinated herds, it would have been interesting to present them, maybe just as descriptive data.
Author Response
Response to Reviewer 2
We thank the reviewer for constructive criticism and useful comments. Please see our responses below.
- Although interesting this paper lacks clarity. Several points in the material and methods needs clarification and results should be better presented in order improve the reader’s understanding.
Response: We are glad that the reviewer finds our paper interesting and have tried to make it more clear by revising it in accordance with the comments received (see below). However, we must point out that reviewer 2 indicated that the results were clearly presented, and the conclusions fully supported by the results.
- The line number are missing, which make the review difficult.
Response: We are sorry for the inconvenience. Omitting line numbers was a mistake when we submitted the original version of the manuscript. Line numbers are however inserted in the updated version.
- You should have your text read by a native speaker. As an example, I believe the sentence “freedom from BRSV” is not correct in English.
Response: We fail to understand that the wording “freedom from BRSV” is incorrect. Is it the word “from” that is wrong? If so, the wording “freedom from disease/illness” is commonly used in scientific texts, for example in the OIE Terrestrial Animal Health Code - 28/06/2019 (“That the internationally recognized 'five freedoms' (freedom from hunger, thirst and malnutrition; freedom from fear and distress; freedom from physical and thermal discomfort; freedom from pain, injury and disease; and freedom to express normal patterns of behavior) provide valuable guidance in animal welfare”). Therefore ”freedom from BRSV/BCoV”, in our opinion, would also be applicable and correct.
Since reviewer 2 considered that English language/style was fine and that only minor spell check is required, we have not sent the manuscript for language editing and leave this issue to the discretion of the editor. However, we would be most grateful if the reviewer could give other examples of incorrect use of the English language so that we can make corrections.
Abstract:
- “BRSV RI herds had significantly higher odds of diarrhea in calves and young stock”: You should clarify compared to what? (this is also a comment latter in your results).
Response: We understand that this quotation is from the Abstract. Since the abstract cannot contain more than 200 words the text is very compressed and has perhaps lost clarity. However, all comparisons are made to FREE herds, but this might have been lost when the text in the Abstract was shortened. We have tried to clarify this in the Abstract by using bullet points (lines 28-36).
This has also been clarified in the Results section (lines -139, 153-156,164-167) and in the caption of Table 2.
Introduction:
- You should add information on the dairy industry in Sweden (economical importance, antimicrobial use regulations (specifically beta-lactams and fluoroquinolones which are classifies as important in human medicine), etc…)
Response: In drafting the manuscript we discussed to include some basic data related to what is suggested by the reviewer and use that to further discuss the results obtained in a wider context. However, the study was designed to specifically investigate the effects of freedom from the infections BRSV and BCoV in a generic manner. Thus, the herds were not selected to represent the national situation in Sweden and therefore, the results obtained, with respect to production, herd size, occurrence of AMR etc., are also not representative for the national situation. To discuss the results in a national context, or in comparison to the situation in other countries, would therefore be irrelevant in our opinion and therefore refrained from giving the detailed information suggested. For the same reason it would be speculative to discuss the results in the context of the national regulations on antibiotics or the use antibiotics. However, if the editor considers that such information would be informative, we can add a paragraph on this in the Introduction. We however think that such a paragraph would rather burden the paper than improve it.
- “All animals born in a herd after an infection has passed will be naïve and susceptible to a new infection”: You should clarify this sentence; it seems in contradiction with the sentence just before.
Response: No there is no contradiction in this. When an infection is active in a herd, most animals get infected and develop antibodies which are detectable for years. However, animals born after the infection has passed and cleared from a herd, will not have been infected. They have therefore not developed antibodies and are thus susceptible if the infection is reintroduced into the herd. This fact is also the basis for the classification of a herd’s infection status by using homebred primiparous cows as sentinels for the presence of infection. If homebred primiparous cows are tested and found antibody negative this shows that the infection has not been active in the herd since these animals were born. However, if primiparous cows are found to be antibody positive the infection has been at some time active in the herd at least during the lifespan of the tested animals. On the contrary, the antibody status of older animals gives no indication of the present infection status because antibodies can be detected for a long period of time after an infection has passed through a herd.
- “that specific antibodies acquired remain detectable for years, even without reinfection” this sentence seems in contradiction with what you say latter in the discussion about short lived immunity. Please clarify. Are the immunity mechanisms the same for both virus, etc?
Response: There is no contradiction. Even though antibodies can be detected in serum and milk for several years after infection with the viruses under study, the animals seem susceptible to reinfections, as has been shown for BCoV [1], [2], and most likely also occurs with BRSV. This phenomenon may be due to that local immunity at the mucosal surface in the gut and/or respiratory epithelium has a much shorter duration than the systemic immune response [3],,[4] or to mechanisms of immune evasion 2. This has been clarified on line 215.
Material and Methods:
- If I am not mistaken, your design is a cohort prospective and not retrospective (you assessed the herds status during the study).
Response: Thank you for this observation. You are right that we followed the herds over time for two years and registered data consecutively during that period. However, the status of the herds were not decided until the end of the study and thus data were used retrospectively. We also used data collected before the start of the study (SOMRS) and therefore would like to call the study an “ambi-directional cohort study” (line 286).
- “they were selected for invitation based on antibody status” could you clarify which status ?
Response: The full sentence is “To maximize the variation in antibody status to BRSV and BCoV of the herds, they were selected for invitation based on antibody status in the most recent nationwide screening of bulk tank milk which was performed in 2013 (A. Ohlson, unpublished data)”. How the selection was performed is then described in the following paragraphs. We understand that separating the description of the selection process on several paragraphs could make it unclear. We have therefore made editorial changes and put the description in a single paragraph (lines 308-321). Hopefully, this clarifies how the selection was done.
- I am not sure why you wanted to “increased the likelihood of recruiting negative farms to the study”.
Response: As both BRSV and BCoV are common endemic diseases in cattle herds in Sweden, the number of FREE herds are limited. In the selection process, we did not want to end up with a set of herds where the vast majority were RI or PSI because this would have made it impossible to evaluate any effects of freedom. We therefore tried to increase the likelihood of recruiting FREE herds by inviting all herds that were FREE in the most recent national screening, as stated in the text.
- Did you consider the sensitivity and susceptibility to your test at the herd level? Indeed, in a large herd where you test for three PPM, you will have less chances of detecting your outcome than in a small herd where you test 3 PPM. (you could at minimum mention it in the discussion).
Response: The within-herd spread of BCoV and BRSV is effective, and all susceptible animals are usually affected within a short period of time and thus the within-herd prevalence of antibodies after an infection is high , [5], [6], [7]. BCoV- and BRSV-specific antibodies assessed from pooled milk samples of home-bred primiparous cows provides a herd indicator for recent virus circulation in both small and large herds. The specificity and sensitivity would be very low for the purpose to identify newly infected herds if using bulk tank milk (BTM) since antibodies are detectable for a long time after infection (many years).
This strategy, testing a few primiparous cows for herd status, is also used in the Norwegian control program for BCoV and BRSV [8] as well as in several published research studies from Norway and Sweden. Such a strategy has also been successfully used to determine BVDV herd-status at a large scale [9]. In very large herds where there are separate isolated animal units, this strategy might not be a true reflection of the overall herd status. Such large-scale herds are, however, not present in this study.
- It is not clear how you establish the number of herds (40 per group) and why in this part of the materiel and method you did not separate RI and PSI whereas you did in the other part of the study. It is also unclear why you chose the prevalence of tetracycline resistance to measure the number of calves you need by herd.
Response: It was difficult to make power calculations for the study because data on the various parameters used in the study were not known and difficult to estimate. However, evaluation of possible effects on AMR was a prime objective of the study, and we had data on the prevalence of tetracycline resistance in E. coli from calves in previous studies. We therefore used that to estimate the number of herds needed. Also, we did not consider that there would be a large difference between RI and PSI herds and therefore did not separate them at this stage.
- Why did you not use also an ESBL/AmpC enrichment?
Response: It would of course have been interesting to include ESBL/AmpC, but the budget did not allow for this because we had decided to quantify occurrence of quinolone resistant and tetracycline resistant E. coli – which already was quite laborious. Also, use of third generation cephalosporins as well as occurrence of ESBL/AmpC resistance is not very common in cattle in Sweden. In a previous study from our group such resistance in E. coli was detected in only 11% of calves and 18% of farms whereas quinolone resistance was detected in 49% of calves and 60% of farms[10]. Since the study was not intended to evaluate occurrence of resistance per se, but instead to investigate if freedom from the endemic infections BRSV and BCoV impacted occurrence of resistance, we chose to focus on quinolone and tetracycline resistance. In cattle practice in Sweden, specifically tetracyclines, but also fluoroquinolones, are much more often used than cephalosporins and resistance to these antibiotics is also more common. We therefore considered that by using them as indicators of resistance we would get wider ranges in the data on prevalence in individual calves which would be beneficial for the statistical analyses. Also, in a previous paper we saw indications that there might be a clonal spread of quinolone resistant E. coli between cattle herds[11] and we wanted to see if this was mitigated by the biosecurity routines in herds free from BRSV/BCoV. The present study also indicated that this might be the case, which is discussed on lines 239-241.
- Could you add some information on antimicrobial use in Sweden? Maybe it would help understanding your choice.
Response: Please see the response to the previous comment above.
- Table 2 : P < 0.5 (did you mean 0.05?). Same remark for the other P.
Response: Thank you for noticing this typo. This has been corrected.
- Could you provide a reference to justify that the small size sample is a good reason not to consider interaction?
Response: We have inserted the reference Leon AC, Heo M. Sample Sizes Required to Detect Interactions between Two Binary Fixed-Effects in a Mixed-Effects Linear Regression Model. Comput Stat Data Anal. 2009;53(3):603-8. Epub 2010/01/20. doi: 10.1016/j.csda.2008.06.010. PubMed PMID: 20084090; PubMed Central PMCID: PMCPMC2678722 on line 454.
Results:
- In the paragraph: 2.4: For the sampling 3: 32%, 33%, 11%, and 25% makes 101%.
Response: This is due to an erroneous rounding of decimal digits. It is now changed to “32%, 33%, 10% and 25%” (line 112).
- In paragraph 2.5: “Quinolone-resistant E. coli (QREC) were isolated from calves on all but one farm” (line 124). This is unusual and should be better explain with antimicrobial use in this country.
Response: It may seem that QREC was very common since they were isolated from almost all farms and also the fact that about half of the calves carried QREC (mean 49%). It should be kept in mind, however, that a selective, and therefore highly sensitive, method was used to isolate QREC. Also, the median for the within-sample prevalence was 0%, indicating that QREC constituted only a small part of the enteric E. coli in the calves that carried QREC. The data is difficult to compare in a proper context since similar studies from other countries are lacking, but in a previous study from our group using the same methodology as in the present study, we isolated QREC from 60% of the farms[12]. In that study only three calves from each herd were sampled which, in comparison to the present study were 18 calves were sampled in each herd, reduced the likelihood of finding at least one calf with QREC in a herd.
It is also noteworthy that 6.5% and 5.2% of the randomly selected E. coli were resistant to nalidixic acid and ciprofloxacin, respectively. This is about the same levels as reported as the mean levels for calves under one year of age reported in the ECDC-EFSA summary report for the EU monitoring of AMR [13]. This indicates that the occurrence of QREC in the Swedish calves is not unusual.
We therefore do not consider the levels of QREC observed in the study remarkably high, which is in agreement with the restricted use of fluoroquinolones in cattle in Sweden as documented in the annual reports from the national monitoring program Swedres-Svarm (https://www.folkhalsomyndigheten.se/publicerat-material/publikationsarkiv/s/swedres-svarm-2019/).
- In the paragraph, 2.6.1: It is not clear to me if you also tested risk factors for the herds that were positive for both BCoV and BRSV as an exposure, if you did not could you tell why.
Response: This is a highly relevant comment, and it would have been interesting to make that evaluation. However, our priority was to evaluate BRSV and BCoV separately. To also consider BRSV-positive and BCoV-positive herds as a separate category would have made that evaluation difficult due to the study’s limited sample-size.
- When you say : “Recently infected herds had significantly higher odds of diarrhea in calves” could you say significantly higher odds than what is it FREE and PSI together or just one of them and if so which one? The same remark could be done for the paragraph 2.6.2.
Response: This has been clarified on lines 155-156 and lines 166-167. See also similar comment and response on Abstract above.
- I do not understand what your final model is. Maybe you should use a table to present these results to clarify.
Response: Different statistical models were used for different outcomes so there is no overall final model. The different models used are presented on lines 449-470 and in Table3. A clarification of this has been included on line 449 (“Different statistical models were used for different outcomes….”)
- I believe you did not present the results of the CFU, why did you perform it?
Response: The CFU count was used to determine the within sample prevalence of QREC and TREC as described on lines 362-390. We have tried to further clarify this on lines 363-365 and 388-390.
Discussion :
- “but due to an overall lower level of biosecurity, hygiene, and management in RI than FREE herds.” It seems to me that FREE herds are supposed to have a higher level of biosecurity. Could you clarify this sentence?
Response: The sentence states that the overall level of biosecurity is lower in RI than in FREE herds. Thus, a higher level in FREE herds than in RI herds. This has been clarified and the sentence now ends “…but due to an overall lower level of biosecurity, hygiene, and management in RI than in FREE herds.” (Line 201).
- And indeed, you say so later in your manuscript: “The lower occurrence of antimicrobial resistance in BRSV FREE herds could therefore be due to an assumed higher level of biosecurity in FREE compared to PSI herds.” “and possibly also mitigate spread of antimicrobial resistance”: you have no data to support this sentence.
Response: No, it is true that we have no data since we did not collect any information on the levels of biosecurity in the herds. However, as stated in the sentence we “assume” that this is the case. We think that this assumption is likely to be true and not overly speculative and therefore interesting to put forward in the Discussion.
- “… therefore supports efforts to establish control programs for BRSV and BCoV in dairy herds”: What do you propose as control programs? Is vaccination a part of it? Although I understand you do not have many data on vaccinated herds, it would have been interesting to present them, maybe just as descriptive data.
Response: This is a very interesting and good question! In Nordic countries we have a different approach compared to other countries. We are free from many infectious diseases, and now we are focusing on BRSV and BCoV as the next step. This study supports our strategy. Control of these viruses could be carried out by vaccination, biosecurity or a combination. Research have shown rapid self-clearance of virus from herds without specific interventions. Molecular epidemiology supports this view—virus varies both temporally and spatially between outbreaks, suggesting that outbreaks are caused by introduction of new virus rather than through reactivation or the existence of carrier animals. This implies that with the current herd size and management conditions in the Nordic countries, herds can self-clear from virus if new introduction is avoided. The Nordic approach for controlling BRSV and BCoV is therefore mainly by biosecurity measures. We also have a long history of eradicating diseases without vaccination. For BCoV and BRSV there are vaccines available in Sweden, but these are mainly used in calf rearing farms and are not commonly used in dairy herds. In the manuscript, line 65-66 this is stated (“Voluntary control programs, based on biosecurity measures, have been suggested in Sweden and are already implemented in Norway”). This is now added also in the discussion for clarity (lines 250-251).
[1] Tråvén M, Thesis, Swedish University of Agricultural Sciences, Uppsala, Sweden, 2000.
[2] Jung HE, Kim TH, Lee HK. Contribution of Dendritic Cells in Protective Immunity against Respiratory Syncytial Virus Infection. Viruses. 2020;12(1). Epub 2020/01/19. doi: 10.3390/v12010102. PubMed PMID: 31952261; PubMed Central PMCID: PMCPMC7020095.
[3] Saif LJ. Bovine respiratory coronavirus. Vet Clin North Am Food Anim Pract. 2010;26(2):349-64. Epub 2010/07/14. doi: 10.1016/j.cvfa.2010.04.005. PubMed PMID: 20619189; PubMed Central PMCID: PMCPMC4094360.
[4] Kimman TG, Westenbrink F, Schreuder BE, Straver PJ. Local and systemic antibody response to bovine respiratory syncytial virus infection and reinfection in calves with and without maternal antibodies. J Clin Microbiol. 1987;25(6):1097-106. Epub 1987/06/01. doi: 10.1128/JCM.25.6.1097-1106.1987. PubMed PMID: 2954996; PubMed Central PMCID: PMCPMC269144.
[5] Alenius S, Niskanen R, Juntti N, Larsson B. Bovine coronavirus as the causative agent of winter dysentery: serological evidence. Acta Vet Scand. 1991;32(2):163-70. Epub 1991/01/01. PubMed PMID: 1666489.
[6] Bidokhti MR, Traven M, Fall N, Emanuelson U, Alenius S. Reduced likelihood of bovine coronavirus and bovine respiratory syncytial virus infection on organic compared to conventional dairy farms. Vet J. 2009;182(3):436-40. Epub 2008/10/07. doi: 10.1016/j.tvjl.2008.08.010. PubMed PMID: 18835795; PubMed Central PMCID: PMCPMC7110579.
[7] Hagglund S, Hjort M, Graham DA, Ohagen P, Tornquist M, Alenius S. A six-year study on respiratory viral infections in a bull testing facility. Vet J. 2007;173(3):585-93. Epub 2006/05/02. doi: 10.1016/j.tvjl.2006.02.010. PubMed PMID: 16647871; PubMed Central PMCID: PMCPMC7110487.
[8] Stokstad M, Klem TB, Myrmel M, Oma VS, Toftaker I, Osteras O, et al. Using Biosecurity Measures to Combat Respiratory Disease in Cattle: The Norwegian Control Program for Bovine Respiratory Syncytial Virus and Bovine Coronavirus. Front Vet Sci. 2020;7:167. Epub 2020/04/23. doi: 10.3389/fvets.2020.00167. PubMed PMID: 32318587; PubMed Central PMCID: PMCPMC7154156.
[9] Lindberg AL, Alenius S. Principles for eradication of bovine viral diarrhoea virus (BVDV) infections in cattle populations. Vet Microbiol. 1999;64(2-3):197-222. Epub 1999/02/24. doi: 10.1016/s0378-1135(98)00270-3. PubMed PMID: 10028173.
[10] Risk factors for antimicrobial resistance in fecal Escherichia coli from preweaned dairy calves. A. Duse, K. P. Waller, U. Emanuelson, H. E. Unnerstad, Y. Persson and B. Bengtsson. J Dairy Sci 2015 Vol. 98 Issue 1 Pages 500-16.
[11] Occurrence and Spread of Quinolone-Resistant Escherichia coli on Dairy Farms. A. Duse, K. Persson Waller, U. Emanuelson, H. Ericsson Unnerstad, Y. Persson and B. Bengtsson. Appl Environ Microbiol 2016 Vol. 82 Issue 13 Pages 3765-73.
[12] Risk factors for antimicrobial resistance in fecal Escherichia coli from preweaned dairy calves. A. Duse, K. P. Waller, U. Emanuelson, H. E. Unnerstad, Y. Persson and B. Bengtsson. J Dairy Sci 2015 Vol. 98 Issue 1 Pages 500-16.
[13] EFSA and ECDC (European Food Safety Authority and European Centre for
Disease Prevention and Control), 2021. The European Union Summary Report on Antimicrobial
Resistance in zoonotic and indicator bacteria from humans, animals and food in 2018/2019. EFSA
Journal 2021;19(4):6490, 179 pp. https://doi.org/10.2903/j.efsa.2021.6490
Reviewer 3 Report
antibiotics-1213211
Overall Comments
The authors provide original research of several critically important disease groups (BRSV, BCoV, and AMR) relevant to the cattle industry of Sweden. The paper could be greatly enhanced if the authors further provided a methods section to expand on the experimental design and analyses used.
Detailed Comments
- Consider rephrasing 2nd paragraph of the introduction: “year after year.” If this is an endemic disease in the cattle population, state that.
- I recognize that the studied diseases are likely chosen by their frequency in the Sweden cattle populations. However, more contextualization on how comorbidity or why researching of these three collective diseases is important.
- Even though it’s purely speculation, why was there not a BCoV effect on milk production? Is there anything in the literature to support this finding?
- The Materials and Methods section needs to be inserted before the results. Specifically, the methods should start with 4.2-3.
- Within the methods section, the authors should indicate more precisely years of study enrollment. Avoid the use of seasons when possible because these are not a standard.
- In addition, the paper could be strengthened if the authors could provide additional context on study enrollment (i.e. geographic regions, cohort selection approach, cohort compensation). Any descriptive statistics on the overall farms would also be useful—are these farms of similar herd sizes and demographics.
- If individual cow IDs were collected, it would be interesting to see repeated measures analyses for the cow models.
- The authors could consider running hierarchical models to further evaluate longitudinal disease trends across time and control for potential aggregations at the farm level.
- It is worth noting (but not over emphasizing) the pros/cons of E. coli as an indicator species. In addition, the microbiota of preweaned calves will not be fully representative of the herd. I realize this might have been a limitation of logistics and sampling, but it’s still worth noting in the discussion.
- “was determined and in addition, the susceptibility to 13 antimicrobials for one randomly selected coli was analysed” – clarification here on the purpose of the randomization would be helpful.
Author Response
Response to Reviewer 3
We thank the reviewer for constructive criticism and useful comments. Please see our responses below.
Overall Comments
The authors provide original research of several critically important disease groups (BRSV, BCoV, and AMR) relevant to the cattle industry of Sweden. The paper could be greatly enhanced if the authors further provided a methods section to expand on the experimental design and analyses used.
Response: There is a Methods section at the end of the manuscript. Due to the specified format of the journal, this section must be placed there, after the sections “Introduction”, “Results” and “Discussion”. Therefore, we cannot place it elsewhere.
Detailed Comments
- Consider rephrasing 2nd paragraph of the introduction: “year after year.” If this is an endemic disease in the cattle population, state that.
Response: The sentence has been rephrased to: “Both diseases are endemic in Swedish cattle herds and it has been suggested that very similar strains of BRSV and BCoV circulate in the cattle population [5, 6] and that specific antibodies acquired remain detectable for years, even without reinfection [9, 10"]” (lines 55-56).
- I recognize that the studied diseases are likely chosen by their frequency in the Sweden cattle populations. However, more contextualization on how comorbidity or why researching of these three collective diseases is important.
Response: We are grateful for this constructive comment and have added the following paragraph as a start of Introduction (lines 42-46): “The burden of disease in food animal production is higher for endemic than for epizootic diseases and they are the major targets for antimicrobial treatments [1]. To prevent and control endemic diseases is therefore likely to be beneficial for productivity and animal welfare but also pivotal for counteracting antimicrobial resistance (AMR) since healthy animals do not need antimicrobial treatments”.
- Even though it’s purely speculation, why was there not a BCoV effect on milk production? Is there anything in the literature to support this finding?
Response: This is an interesting point and briefly already discussed on lines 182-187. A decrease in milk yield is commonly reported from farmers’ experiences as well as from description of outbreaks of BCoV in the literature. The decreased milk yield associated with BCoV, even if large in magnitude (on average 22%), is often described as being transient (<2 weeks[1]). Therefore, such an effect would have been difficult to detect using test-day records, which are usually performed once monthly. No information on actual dates of possible outbreaks were available, and a more detailed analysis could therefore not be performed. Also, it is likely that not all cows in the RI herds were susceptible. The older cows might be antibody positive, and a reinfection is described as mild or with no symptoms at all.
- The Materials and Methods section needs to be inserted before the results. Specifically, the methods should start with 4.2-3.
Response: Please see comment 1. We agree with the reviewer that it would have been better to place the Methods section before “Results”, but we must follow the instructions of the journal and place it here.
- Within the methods section, the authors should indicate more precisely years of study enrollment. Avoid the use of seasons when possible because these are not a standard.
Response: This has been changed to ”Each herd submitted milk samples three times: 1st at enrollment (October 2015 - March 2016); 2nd about one year later (November 2016 - June 2017); 3rd about two years after the first (November 2017 - June 2018)” (lines 288-292). This was also changed on lines 80-81 and on line 308.
- In addition, the paper could be strengthened if the authors could provide additional context on study enrollment (i.e., geographic regions, cohort selection approach, cohort compensation). Any descriptive statistics on the overall farms would also be useful—are these farms of similar herd sizes and demographics.
Response: On lines 80–88 there is some information on the geographical location and size of the herds. In drafting the manuscript, we discussed to include some basic data related to what is suggested by the reviewer and use that to further discuss the results obtained in a wider context. However, the study was designed to specifically investigate the effects of freedom from the infections BRSV and BCoV in a generic manner. Thus, the herds were not selected to represent the national situation in Sweden and therefore, the results obtained, with respect to production, herd size etc., are also not representative for the national situation. To discuss the results in a national context, or in comparison to the situation in other countries, would therefore be irrelevant in our opinion and therefore we refrained from giving the detailed information suggested. However, if the editor considers that such information would be informative, we can add a paragraph on this in the Introduction. We, however, think that such a paragraph would rather burden the paper than improve it.
- If individual cow IDs were collected, it would be interesting to see repeated measures analyses for the cow models.
Response: We had individual IDs from all cows present in the study herds during the study period which enabled us to study repeated data from individual cows. We accounted for correlations between repeated measurements from the same cow using a compound symmetry covariance matrix. This information has been added to the text on lines 463-464 to clarify.
- The authors could consider running hierarchical models to further evaluate longitudinal disease trends across time and control for potential aggregations at the farm level.
Response: We think that evaluating longitudinal disease trends would not be appropriate for this study where data on diseases were collected for only two years. Proper evaluation of trends would in our opinion demand a longer series of registrations than what is available. Also, we are not quite sure that we understand what the reviewer means by ”aggregations at the farm level”.
- It is worth noting (but not over emphasizing) the pros/cons of E. coli as an indicator species. In addition, the microbiota of preweaned calves will not be fully representative of the herd. I realize this might have been a limitation of logistics and sampling, but it is still worth noting in the discussion.
Response: We agree that there are pros and cons in using E. coli from calves as indicator of AMR. However, it is a well-established, feasible and cost-effective way to obtain harmonized data on occurrence of AMR in a population and commonly used – although not perfect. We have added the following sentences on lines 270-274 to put the issue in some perspective. “As outcome indicator of AMR, we used the antimicrobial susceptibility of enteric E. coli from calves. This indicator is commonly used to monitor AMR in farm animals and considered to reflect the selection pressure from use of antimicrobials in animal populations [27]”. Although the indicator does not reflect the complete resistome of the animals, it is a simple and well-established way to evaluate occurrence of AMR in farm animals.”
10 “was determined and in addition, the susceptibility to 13 antimicrobials for one randomly selected coli was analysed” – clarification here on the purpose of the randomization would be helpful.
Response: In programs monitoring resistance in enteric E. coli from animals, for example the program run by EFSA, the Swedish program Svarm and other programs, it is common practice to subculture one colony from the primary culture. Thereby an “isolate” is obtained. The isolate is subsequently tested for antimicrobial susceptibility. Of course, this is not a “randomized” procedure in the strict statistical sense and to avoid ambiguities we have rephrased this, see lines 364-3365 and lines 391-392.
[1] Tråvén M, Thesis, Swedish University of Agricultural Sciences, Uppsala, Sweden, 2000.
Round 2
Reviewer 1 Report
The authors have adequately adressed my major concerns.
Reviewer 2 Report
The authors have answered the questions.
Reviewer 3 Report
Thank you for taking the time to address my comments!
In regards to Comment #8: the authors could perhaps conduct a follow up study for an additional paper to look at temporal relationships across farms. In terms of "aggregations at the farm level” -- I was referring to evaluating AMR outcomes at the farm level instead of individual cattle level. There are some epidemiological assumptions here (i.e. cattle have shared colonization of resistant bacteria, equal exposure, same AB feed, etc. etc.). However, this approach could be informative to look at the population scale and compare farming practices. Just a thought for a future paper.